# Probing the Propeller Regime with Symbiotic X-ray Binaries

**Marina D. Afonina [1,2]** and **Sergei B. Popov [3,*]**

1 Sternberg Astronomical Institute, Lomonosov Moscow State University, Universitetskij pr. 13, 119234 Moscow, Russia; afonina.md19@physics.msu.ru
2 Department of Physics, Lomonosov Moscow State University, 1/2 Leninskie Gory, 119991 Moscow, Russia
3 ICTP—Abdus Salam International Center for Theoretical Physics, Strada Costiera 11, I-34151 Trieste, Italy
* Correspondence: sergepolar@gmail.com; Tel.: +39-040-2240-397

**Abstract:** At the moment, there are two neutron star X-ray binaries with massive red supergiants as donors. Recently, De et al. (2023) proposed that the system SWIFT J0850.8-4219 contains a neutron star at the propeller stage. We study this possibility by applying various models of propeller spin-down. We demonstrate that the duration of the propeller stage is very sensitive to the regime of rotational losses. Only in the case of a relatively slow propeller model proposed by Davies and Pringle in 1981, the duration of the propeller is long enough to provide a significant probability to observe the system at this stage. Future determination of the system parameters (orbital and spin periods, magnetic field of the compact object, etc.) will allow putting strong constraints on the propeller behavior.

**Keywords:** neutron stars; X-ray binaries; accretion





## 1. Introduction

Thousands of neutron stars (NSs) are known as sources of various natures [1–3]. Observational appearance of an NS depends on its parameters (rotation, mass, magnetic field, etc.) and its interaction with the surrounding medium. Some regimes of the interaction of an NS with external matter are not well studied. In particular, this is true for the propeller regime proposed in 1970 by Shvartsman [4]. At this stage, a rapidly rotating magnetosphere prevents the accretion of the gravitationally captured plasma onto the NS surface.

Already in the early 1970s basic properties of the propeller stage and its importance for the evolution of X-ray binaries were well understood [5–8]. However, the direct proof of the existence of this phase of NS evolution was absent. The propeller phase of an NS evolution is an elusive one as the energy release can be rather low and the duration of this stage can be relatively short. Still, it is expected that some X-ray binaries can contain NSs at this stage (see e.g., ref. [9] and references therein). Probably, in several cases of millisecond pulsars [10], standard X-ray pulsars [11], accreting non-pulsating NSs [12,13], and even ultra-luminous X-ray sources [14,15] it is possible to detect transitions from accretion to the propeller stage (and back) by detection of rapid changes in luminosity and spectral properties. Identification of an NS in the propeller regime in a well-studied binary can provide important clues for a better understanding of the interaction between the NS magnetic field and ionized matter.

At the moment, mainly due to X-ray observations, many hundreds of NSs are identified in interacting binary systems with different types of companions, see [16] for a review. For understanding the propeller regime, so-called high-mass X-ray binaries (HMXBs) are especially interesting. There are hundreds of X-ray binaries of this type [17,18]. For our purposes, systems where the compact object captures matter from a slow stellar wind of its companion, are the most important. For some combinations of NS parameters (spin period and magnetic field), parameters of the binary (semi-major axis, eccentricity), and the donor (stellar mass, mass loss rate, and velocity of the stellar wind) the rotating magnetosphere

can expel the captured matter not allowing stable accretion. Among various sub-classes of HMXBs, so-called symbiotic X-ray binaries (SyXBs) can be one of the best candidates to host a propeller.

SyXBs are studied in many papers, see e.g., introductory part in [19] for a brief review. They consist of an NS and a donor at the red giant branch (RGB) or the asymptotic giant branch. Mostly, donors are low-mass late-type giants. Systems are not very numerous as they have a short lifetime determined by the duration of the evolution of the donor. Yungelson et al. [19] estimated that there are less than 40–50 SyXBs in the Galaxy. Still, there might be a few systems with massive donors, i.e., with red supergiants.

Up to recent times, there was just one known X-ray binary with an accreting NS and a red supergiant companion. This is the system 4U 1954+31 [20]. The mass of the donor is estimated to be 7–15 $M_\odot$. The NS in this system has a peculiarly long spin period—about five hours. Such a long spin period can be explained, for example, assuming that the NS has a magnetar-scale magnetic field [21].

Recently, the second SyXB with a supergiant—SWIFT J0850.8-4219—was identified [22]. In this system, the donor is a red supergiant with $T_{\mathrm{eff}} = 3820 \pm 100$ K and mass $\sim$10–20 $M_\odot$. The spin period of the NS as well as the orbital period are not known. The semi-major axis is at least $\gtrsim$300 $R_\odot$ as there is no Roche lobe overflow in this system (the lower limit depends on the size of the donor and probably it is a few times larger than the given estimate). The X-ray luminosity is $(4 \pm 1) \times 10^{35}$ erg s$^{-1}$ and the spectrum $N(E) \propto E^{-\Gamma}$ is rather hard with the photon index $\Gamma < 1$. These properties led the authors to the conclusion that the NS can be at the propeller stage. Then a small fraction of the captured matter (not more than a few percent) still can accrete onto the surface of the NS producing the observed emission.

If the hypothesis proposed by De et al. [22] is correct then the system SWIFT J0850.8-4219 provides a unique opportunity to probe the physics of the propeller regime. In this paper, we analyze evolution under the conditions measured for SWIFT J0850.8-4219 by applying several models of propeller spin-down.

In the next section, we present the basics of the model we use. Section 3 contains a detailed description of various models of the propeller regime proposed by different authors. Then in Section 4, results of the modeling are described. In Section 5, we discuss some of our assumptions and present calculations for alternative assumptions. In the Section 6, we present our conclusions.

## 2. Model

In this section, we describe the basics of the model we use, except for the part related to various approaches to specifying an NS spin evolution at the propeller stage, which is presented in a separate section. At first, we describe the stellar wind model and the calculations of the accretion rate, $\dot{M}$. Then we demonstrate how the spin evolution of the NS is calculated in our model.

### 2.1. Stellar Wind

Stellar wind parameters are extremely important for our modeling. Unfortunately, there are many unsolved problems related to the wind properties at different phases of the evolution of massive stars, see e.g., ref. [23] for a detailed review.

We need to specify stellar parameters throughout the evolution of the donor: from the Main sequence to the supergiant stage. To reach this goal, we utilize PARSEC evolutionary tracks [24]. We use the track for the single mass value $M_* = 14\,M_\odot$ and solar metallicity. For now, we assume that the mass of the NS progenitor at the Zero Age Main sequence is $\sim$30 $M_\odot$ which is important to estimate the age of the NS. The PARSEC track for the 30 $M_\odot$ ends at 6.3 Myr with the onset of helium burning. We assume that the NS is born at 7 Myr after the system formation. Our results are not sensitive to the exact choice of this number in reasonable limits defined by a possible progenitor mass. Everywhere below we will use the time elapsed from the birth of the NS, which is calculated as $t = \mathrm{age}(14\,M_\odot) - 7$ Myr.

Here the age corresponds to the moment of time for which we take stellar parameters from the evolutionary track for the 14 $M_\odot$ star.

In our modeling, we use the following stellar parameters taken from the track: mass $M_*(\text{age})$, radius $R_*(\text{age})$, mass loss rate $\dot{M}_w(\text{age})$, and effective temperature $T(\text{age})$.

At the beginning of the red supergiant (RSG) stage (i.e., the star has an inert helium core and an expanded hydrogen envelope), the second component reaches the maximum radius of 740 $R_\odot$, then decreases to 500 $R_\odot$ and starts to increase again at the latest stages of evolution. If the donor star fills its Roche lobe then a stream of gas flows through the inner Lagrange point. This is not the case for the system SWIFT J0850.8-4219. Therefore, we can find the minimum separation $a$ of a circular orbit using the value 740 $R_\odot$ for the maximum radius of the donor. This value is not exceeded until the second component starts to expand after the exhaustion of helium in the core (we do not consider further evolution of the donor). From [25] we get the ratio of the effective radius of a Roche lobe $R_L$ and the semi-major axis $a$:

$$\frac{R_L}{a} = \frac{0.49q^{2/3}}{0.6q^{2/3} + \ln(1 + q^{1/3})},$$
(1)

where $q$ is the mass ratio. For the second component $q = M_*/M = 10$, where $M = 1.4\,M_\odot$ is the NS mass. If $R_L = 740\,R_\odot$ then the tightest possible orbit is $a = 1280\,R_\odot$.

The track is terminated when $R_*$ reaches 740 $R_\odot$ again. Thus, during the evolution of the binary, the second component never fills its Roche lobe.

With the mass, radius, and temperature known, we calculate the wind velocity following the prescription from [26]. The evolution of the mass loss rate $\dot{M}_w$ and the wind velocity $v_w$ near $r = a = 1280\,R_\odot$ are shown in Figure 1.

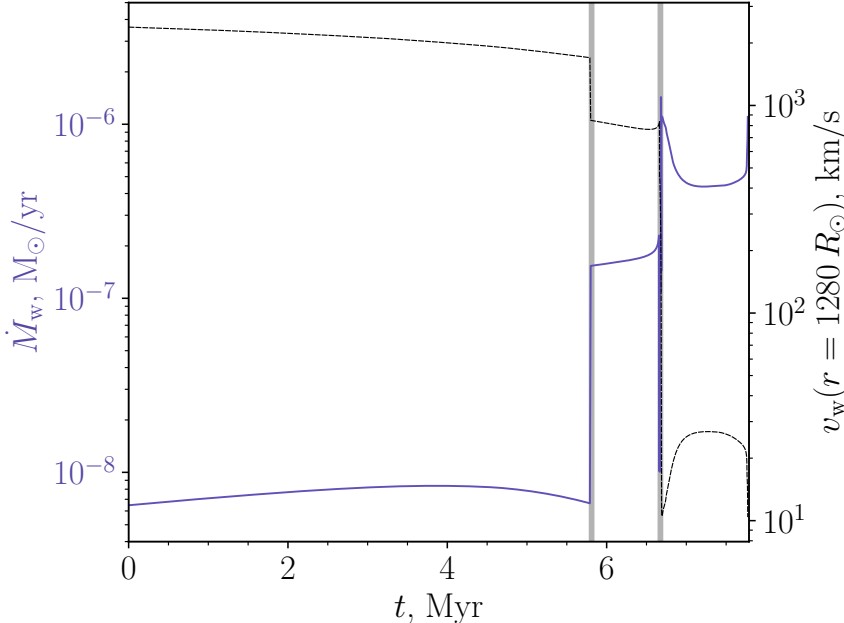

**Figure 1.** Donor mass loss rate $\dot{M}_w$ (blue solid curve) and wind velocity $v_w(r)$ (black dashed curve) at $r = 1280\,R_\odot$ over the time elapsed since the birth of the NS. Vertical lines indicate bi-stability jumps. The second bi-stability jump determines the change in donor evolution from the Main sequence to the red supergiant stage.

We now present a detailed description of our $v_w$ calculations. The dependence of the wind velocity $v_w$ on the distance from the center of the donor $r$ is given by a beta-type velocity law:

$$v_w(r) = v_\infty \left(1 - \frac{R_*}{r}\right)^\beta,$$
(2)

where $v_\infty$ is the terminal velocity.

The terminal velocity can be expressed in terms of the escape velocity near the stellar surface $v_{esc} = \sqrt{2GM_*/R_*}$. The ratio $v_\infty/v_{esc}$ depends on the evolutionary stage of the donor. We can distinguish three stages. Each subsequent stage begins with a sharp jump in $\dot{M}_w$. The authors of the paper [27] suggest that the jump in mass loss is closely related to the bi-stability jump, which is the observed decrease in the ratio $v_\infty/v_{esc}$.

The first bi-stability jump corresponds to a decrease in the effective temperature of the donor, $T$, down to 23,000 K during the evolution on the Main sequence. In this case, Fe IV recombines to Fe III in the inner part of the wind. This makes the wind acceleration more effective, increases $\dot{M}_w$, but leads to a decrease in $v_\infty/v_{esc}$ from 2.6 to 1.3. According to the assumption from [26], the second bi-stability jump is also explained by the recombination of Fe III to Fe II. This takes place at $T \sim 10,000$ K and reduces $v_\infty/v_{esc}$ from 1.3 to 0.7. For the 14 $M_\odot$ donor we expect this transition to occur during the sharp drop in effective temperature on the way to the RSG phase.

The parameter $\beta$ is about unity for O- and B-stars [26]. So, we adopt $\beta = 1$ for the first two stages. Now we determine the value of $\beta$ value at the RSG stage. The SWIFT J0850.8-4219 counterpart is a K3-K5 type RSG. The dependence of the ratio $v_w/v_\infty$ for K4 type RSGs is shown in [28]. For now, we are interested in $r < 3R_*$, where $\beta \approx 2 \div 3$. We take $\beta = 2$ for the RSG stage.

For the NS in the binary with the semi-major axis $a$, we calculate the rate of matter capturing, $\dot{M}$, which is the NS accretion rate if accretion is possible. According to Bondi [29], for an NS surrounded by a stellar wind with the density $\rho_w$ the accretion rate is $\dot{M} \propto (GM)^2\rho_w v^{-3}$, where $v$ is the NS velocity relative to the wind. In [22] the authors use the following equation:

$$\dot{M} \approx 2.5\pi(GM)^2\rho_w(a)v^{-3}(a).$$
(3)

The coefficient 2.5 is in correspondence with numerical modeling and analytical estimates for moderate Mach numbers that are expected in the case of SWIFT J0850.8-421, see [30,31].

From the continuity equation $\dot{M}_w = 4\pi a^2\rho_w(a)v_w(a)$ we get $\rho_w(a)$. Here the relative velocity $v \approx \sqrt{v_w^2 + v_K^2}$ includes the wind velocity $v_w$ and the sum of the Keplerian velocities of both stars $v_K = \sqrt{G(M_* + M)/a}$. Here we neglect the sound velocity as it is expected to be lower than the sum of the wind and orbital velocities. Finally, we obtain

$$\dot{M} = \frac{5}{8}\frac{(GM)^2}{a^2 v_w v^3}\dot{M}_w.$$
(4)

Using $\dot{M}_w$ and calculating $v_w$ from the properties of the donor star, we can find $\dot{M}(t)$.

*2.2. Spin Evolution of NSs*

In our description of the spin evolution, we basically follow the standard approach presented e.g., in the book [32]. A more recent description can be found in the review [33].

The evolutionary status of an NS depends on the interplay between the NS spin period, $P$, its magnetic field $B$, and the parameters of the surrounding medium. It is convenient to describe the latter in terms of the mass accretion rate, $\dot{M}$, even if there is no accretion onto the surface.

The initial spin period in our main calculations is assumed to be $P_0 = 100$ ms. This is in correspondence with usual estimates of typical NS parameters, see e.g., [34,35]. We perform calculations for different values of the magnetic field. However, as we focus on the properties of SWIFT J0850.8-4219, we are mainly interested in the values $\gtrsim 10^{12}$ G (see the field estimate in [22]).

We do not include in our calculations magnetic field decay (see [36] for a review). This is justified by three considerations. On the one hand, the NS in SWIFT J0850.8-4219 cannot be older than $\sim 10^7$ yrs as it has a massive companion. Thus, we can safely ignore a long-term field evolution due to Ohmic dissipation in the crust. On the other hand, the NS

might not be very young (e.g., due to the absence of a supernova remnant). Then, in the case of standard pulsar-like fields, we can ignore a possible early episode of field decay suggested in [37]. Also, we can neglect the early rapid evolution of a magnetar-scale field. It is expected that after a few e-folding times such evolution saturates when the compact object reaches the so-called Hall attractor stage [38]. If such early rapid episodes of field evolution took place in the life of the NS in SWIFT J0850.8-4219, then on the scale of our model it just modifies the assumption about the initial spin period. Finally, we can neglect field decay due to accretion, see e.g., [39], as in the case of SWIFT J0850.8-4219 we are dealing with a young system with a massive donor not overfilling its Roche lobe, so the total amount of accreted matter (as well the duration of accretion) cannot be sufficiently high to influence the magnetic field significantly.

The NS is assumed to have constant mass $M = 1.4\,M_\odot$ and the moment of inertia $I = 10^{45}$ g cm$^2$.

We distinguish four main evolutionary stages: ejector, propeller, accretor, and georotator (see [32,33]). They can be characterized by ratios between some characteristic radii. They are the light cylinder radius $R_\mathrm{l}$, gravitational capture radius $R_\mathrm{G}$, corotation radius $R_\mathrm{co}$, magnetospheric radius $R_\mathrm{m}$, and Shvartsman radius $R_\mathrm{Sh}$.

The light cylinder radius is:

$$R_\mathrm{l} = c/\omega, \tag{5}$$

where $c$ is the velocity of light and $\omega = 2\pi/P$ is the spin frequency.

The corotation radius is:

$$R_\mathrm{co} = (GM/\omega^2)^{1/3}. \tag{6}$$

Recently, Lyutikov [40] proposed a modification to the standard approach to calculate the centrifugal barrier. We discuss this possibility in Section 5.

The gravitational (aka Bondi) radius:

$$R_\mathrm{G} = \frac{2GM}{v^2}. \tag{7}$$

Here $G$ is the Newton constant, and $v$ is the velocity relative to the medium.

Shvartsman radius:

$$R_\mathrm{Sh} = \left( \frac{2\mu^2 (GM)^2 \omega^4}{\dot{M} v^5 c^4} \right)^{1/2}. \tag{8}$$

Here $\mu = BR^3$ is the magnetic moment, $R = 10$ km is the NS radius.

Alfvén radius:

$$R_A = \left( \frac{\mu^2}{2\dot{M}\sqrt{2GM}} \right)^{2/7}. \tag{9}$$

At the ejector stage, the external matter (the stellar wind from the second component) is stopped at the Shvartsman radius $R_\mathrm{Sh}$, which is greater than both the light cylinder radius $R_\mathrm{l}$, and the gravitational capture radius $R_\mathrm{G}$. As the NS spins down, the characteristic radius, at which the outer matter is stopped, $R_\mathrm{Sh}$, decreases until either it reaches the light cylinder radius $R_\mathrm{Sh} = R_\mathrm{l}$, or the external matter becomes gravitationally captured when $R_\mathrm{Sh} = R_\mathrm{G}$. At the subsequent evolutionary phase—propeller,—external matter penetrates inside the light cylinder. Thus, the relativistic particle wind is terminated. However, the matter cannot accrete, yet, as it is stopped by a rapidly rotating magnetosphere at $R_\mathrm{m}$. Finally, if the magnetospheric radius $R_\mathrm{m}$ is smaller than the corotation radius $R_\mathrm{co}$, then accretion onto the NS surface is allowed.

At all stages, the spin period of the NS changes due to external torques. The Euler equation for the spin period evolution can be written as:

$$\dot{P} = \frac{P^2}{2\pi I} K. \tag{10}$$

Here $K$ is the external (decelerating or accelerating) moment.

The decelerating moment for the ejector stage is defined as:

$$K_{\mathrm{E}} = 2\frac{\mu^2}{R_{\mathrm{l}}^3}.$$ (11)

In general, we assume that the ejector-propeller transition corresponds to the equality $R_{\mathrm{Sh}} = R_{\mathrm{G}}$. However, for high values of $v$, $R_{\mathrm{G}} \propto v^{-2}$ can be less than $R_{\mathrm{l}}$. So the transition condition is $R_{\mathrm{Sh}} = R_{\mathrm{l}}$. Then for the critical period we have:

$$P_{\mathrm{EP}} = \begin{cases} \dfrac{2\pi}{c}\left(\dfrac{2\mu^2}{4\dot{M}v}\right)^{1/4}, & R_{\mathrm{G}} > R_{\mathrm{l}} \\[2ex] \dfrac{2\pi}{c}\left(\dfrac{2\mu^2(GM)^2}{\dot{M}v^5}\right)^{1/6}, & R_{\mathrm{G}} \leq R_{\mathrm{l}}. \end{cases}$$ (12)

After this transition the external matter can interact with the NS magnetosphere. The magnetospheric radius $R_{\mathrm{m}}$ is always kept smaller than $R_{\mathrm{l}}$, since the existence of a magnetosphere outside of the light cylinder is not possible.

At the accretion stage, in addition to the decelerating moment ($K_{\mathrm{sd}}$), an accelerating one ($K_{\mathrm{su}}$) appears. Thus, we have: $K_{\mathrm{A}} = K_{\mathrm{sd}} - K_{\mathrm{su}}$. The accelerating term is different for the cases of spherical and disc accretion. For the spin-down and spin-up torques, we can write:

$$K_{\mathrm{sd}} = k_{\mathrm{t}}\frac{\mu^2}{R_{\mathrm{co}}^3},$$ (13)

where $k_{\mathrm{t}}$ is a dimensionless coefficient $\sim 1$, and

$$K_{\mathrm{su}} = \begin{cases} \dot{M}\sqrt{GMR_{\mathrm{A}}}, & \text{disc} \\ \dot{M}\eta\Omega R_{\mathrm{G}}^2, & \text{no disc}. \end{cases}$$ (14)

Here $\Omega = \sqrt{G(M_* + M)/a^3}$, and we assume $\eta = 1/4$.

An accretion disc is formed if the specific angular momentum of the accreting matter is larger than the Keplerian momentum at the magnetospheric radius. This corresponds to the condition:

$$\sqrt{GMR_{\mathrm{m}}} \leq \eta\Omega R_{\mathrm{G}}^2.$$ (15)

At the stage of accretion, spin-up and spin-down might balance each other and so an equilibrium is reached. The equilibrium period is obtained from the condition $K_{\mathrm{su}} = K_{\mathrm{sd}}$. The radius of the magnetosphere does not depend on the period at the accretion stage, so $P_{\mathrm{eq}}$ can be written in the form:

$$P_{\mathrm{eq}} = 2\pi\mu\sqrt{\frac{k_{\mathrm{t}}}{GMK_{\mathrm{su}}}}.$$ (16)

To specify the exact value of $k_{\mathrm{t}}$ we proceed in the following way. In our case, accretion proceeds with a disc formation. The inner radius of the disc is $R_{\mathrm{d}} = fR_{\mathrm{A}}$, $f \sim 0.5$–$1.0$, e.g., ref. [41]. One can define the so-called fastness parameter: $\omega_{\mathrm{crit}} = (R_{\mathrm{d}}/R_{\mathrm{co}})^{3/2}$. Numerical calculations, e.g., [42], demonstrate that $\omega_{\mathrm{crit}}$ is such that the ratio $R_{\mathrm{d}}/R_{\mathrm{co}}$ is from $\approx 0.9$ up to $\lesssim 1$. We use the mean value 0.96. Then, from

$$k_{\mathrm{t}}\frac{\mu^2}{R_{\mathrm{co}}^3} = \dot{M}\sqrt{GMR_{\mathrm{A}}}$$ (17)

for $f = 1$ we obtain $k_{\mathrm{t}} = (1/0.96)^3/(2\sqrt{2}) \approx 0.4$.

After its birth, the NS with $P_0 = 0.1$ s and $B \gtrsim 10^9$ G appears at the ejector stage as the second star in the binary is on the Main sequence and does not produce a strong wind. The NS then gradually spins down while the rate of mass loss from the companion due to the stellar wind increases. The NS can make transitions from ejector to propeller, and then to

accretor. In the next section, we present different propeller models used in our study, and then in Section 4, we describe the whole modeled evolution.

## 3. Propeller Stage

In our study, we use several propeller models to see if any of them provide a significant probability of capturing a system like SWIFT J0850.8-4219 at this stage.

At the propeller stage, the external matter is stopped at $R_{\rm m}$ and $R_{\rm m} > R_{\rm co}$. So, the accretion of large amounts of matter is prevented by the centrifugal barrier. Around the magnetosphere of the NS, an envelope of external material extends up to $R_{\rm G}$. Various propeller models are characterized by a density profile of this envelope which depends on the energy release and transfer. Thus, pressure outside the magnetospheric boundary can be different in different models. This leads to different expressions for $R_{\rm m}$. Additionally, as each propeller model corresponds to a different regime of the interaction between the magnetosphere and the envelope, it results in different spin-down torques $K$. The basic properties—decelerating moment and magnetospheric radius—of all the models and the corresponding references are presented in Table 1. A detailed review of the various propeller models can be found in [33].

We apply five models of the propeller stage, denoted A, A1, B, C, and D. Models A and A1 are based on the paper [7]. The difference between the two models is related to different definitions of the magnetospheric radius. In A1 we follow the original proposal by Shakura. In this case, $R_{\rm m}$ depends on the spin period. In model A we use the constant value $R_{\rm m} = R_{\rm A}^{7/9} R_{\rm G}^{2/9}$ as in all other models (B, C, D). Thus, for A1 and other models, we have different values of the critical period for the propeller-accretor transition.

The propeller-accretor transition period for models A, B, C, D (for $R_{\rm m} < R_{\rm G}$):

$$P_{\rm PA} = \pi \left( \frac{\mu^2}{2\sqrt{2}GM\dot{M}v^2} \right)^{1/3}. \tag{18}$$

For model A1 (for $R_{\rm m} < R_{\rm G}$):

$$P_{\rm PA} = \frac{2\pi}{(GM)^{5/7}} \left( \frac{\mu^2}{\sqrt{2}\dot{M}} \right)^{3/7}. \tag{19}$$

We do not consider the subsonic propeller stage (see [43] about it), assuming that accretion starts since $R_{\rm m} < R_{\rm co}$. Probably, soon after the accretion starts, it can resemble the settling accretion regime [44] even for relatively large inflow rates while the envelope around the magnetosphere is cooling down.

At the propeller stage, it is necessary to consider separately the cases when $R_{\rm m} < R_{\rm G}$ and $R_{\rm m} > R_{\rm G}$ because the magnetospheric radius is calculated differently in these cases. We control this carefully, because depending on the mass loss rate of the donor, one or the other condition can be realised in a wide range of magnetic fields. Therefore, in the Table 1 we provide formulae for both cases. However, we note that for realistic parameters, there is no transition from the propeller to the accretor stage for $R_{\rm m} > R_{\rm G}$, i.e., there are no georotators. Therefore we do not provide the critical period for this case.

The decelerating moment is defined differently in the considered models, see Table 1. In some cases, it depends on the additional parameter—free-fall velocity at the magnetospheric radius: $v_{\rm ff}(R_{\rm m}) = \sqrt{2GM/R_{\rm m}}$.

Among all the considered models, model A provides the most rapid spin-down as the magnetospheric radius stays constant and the period grows exponentially. The slowest evolution is the property of the model D. In this case, the spin-down rate $\dot{P}$ can be even slower than at the stage of ejection. Francischelli and Wijers [45] provided some argumentation against the fastest variants of the propeller spin-down. Nevertheless, here we present results for various possibilities.

**Table 1.** Propeller models and corresponding values of the decelerating torque $K$ and the magnetosphere radius $R_{\mathrm{m}}$.

| Model | Authors | $K = -I\dot{\omega}$ | $R_{\mathrm{m}}, R_{\mathrm{m}} < R_{\mathrm{G}}$ | $R_{\mathrm{m}}, R_{\mathrm{m}} > R_{\mathrm{G}}$ |
|---|---|---|---|---|
| A1 | Shakura (1975) [7] | $\dot{M}\omega R_{\mathrm{m}}^2$ | $(\mu^2 v R_{\mathrm{G}}^{1/2}/(2\dot{M}\omega^2))^{2/13}$ | $(\mu^2 v R_{\mathrm{G}}^2/(2\dot{M}\omega^2))^{1/8}$ |
| A | Shakura (1975) [7] | $\dot{M}\omega R_{\mathrm{m}}^2$ | | |
| B | D & O (1973) [6] | $\dot{M}\sqrt{2GMR_{\mathrm{m}}}$ | $R_{\mathrm{A}}^{7/9} R_{\mathrm{G}}^{2/9}$ | $(\mu^2 v R_{\mathrm{G}}^2/(2\dot{M}v))^{1/6}$ |
| C | I & S (1975) [8] | $\dot{M}v_{\mathrm{ff}}^2/(2\omega)$ | | |
| D | D & P (1981) [43] | $\dot{M}v^2/(2\omega)$ | | |

## 4. Results

In this section we present the results of calculations of the evolution of an NS in a binary system, taking into account the evolution of the donor.

### 4.1. Detailed Spin Evolution of a Typical Neutron Star in a Binary

We show the results for an NS with $P_0 = 100$ ms in a binary with the semi-major axis $a = 1280\,R_\odot$. The eccentricity is assumed to be zero throughout the evolution. The results for the standard magnetic field $B = 10^{12}$ G are shown in Figure 2 and for the higher value $B = 4 \times 10^{12}$ G in Figure 3. The top panels show the evolution of $\dot{M}$ and the middle and bottom panels show the absolute value of the period derivative $|\dot{P}|$ and the period $P$ over the NS age $t$. The panels to the right are the zoomed region where most of the transitions occur as the donor approaches the RGB.

In Figure 2 we see that for all models (except model D) the propeller stage is very brief. It is therefore highly improbable to detect one of just two known systems with RSG donors at this evolutionary phase.

The NS evolution at the ejector stage and the transition to the propeller stage are independent of the propeller model and are common to all NSs with the same $P_0$ and $B$. At this stage the NS is slowing down, so the period derivative is positive. From Equation (11), $\dot{P} \propto P^{-3}$, independent of $\dot{M}$ and $v$, and decreases with time until the transition to the propeller.

The time of the ejector-propeller transition is different for the two magnetic field values. As $\dot{P} \propto B^2$, the NS with the higher field evolves faster. On the other hand, $P_{\mathrm{EP}} \propto B^{1/2}$. So, for a larger field, it is necessary to reach a larger spin period to become a propeller. Finally, under these external conditions $P_{\mathrm{EP}} \propto (\dot{M}v)^{-1/4}$ and decreases over time with the donor evolution, because the quantity $\dot{M}v$ increases until the donor reaches the RGB. It is difficult to predict the outcome of calculations with these three dependencies. It is therefore necessary to calculate the evolution numerically.

The NS with $B = 4 \times 10^{12}$ G reaches the propeller stage at $\approx 6$ Myr, before the donor becomes an RSG. So, this transition is mainly driven by the spin evolution. For the lower field $B = 10^{12}$ G the transition occurs when the mass loss by the secondary component is drastically increased. At this moment, the transition from the ejector is inevitable. With the peak of $\dot{M}$ at $\sim 6.7$ Myr, the transition period $P_{\mathrm{EP}}$ reaches its minimum of $\sim 100\,B_{12}^{1/2}$ ms, where $B_{12} = B/10^{12}$ G. At the same time, from the integration of the Euler equation (Equation (10)) with $K_{\mathrm{E}}$, the period reached by 6.7 Myr is $\gtrsim 1\,B_{12}$ s, which is sufficient for the transition.

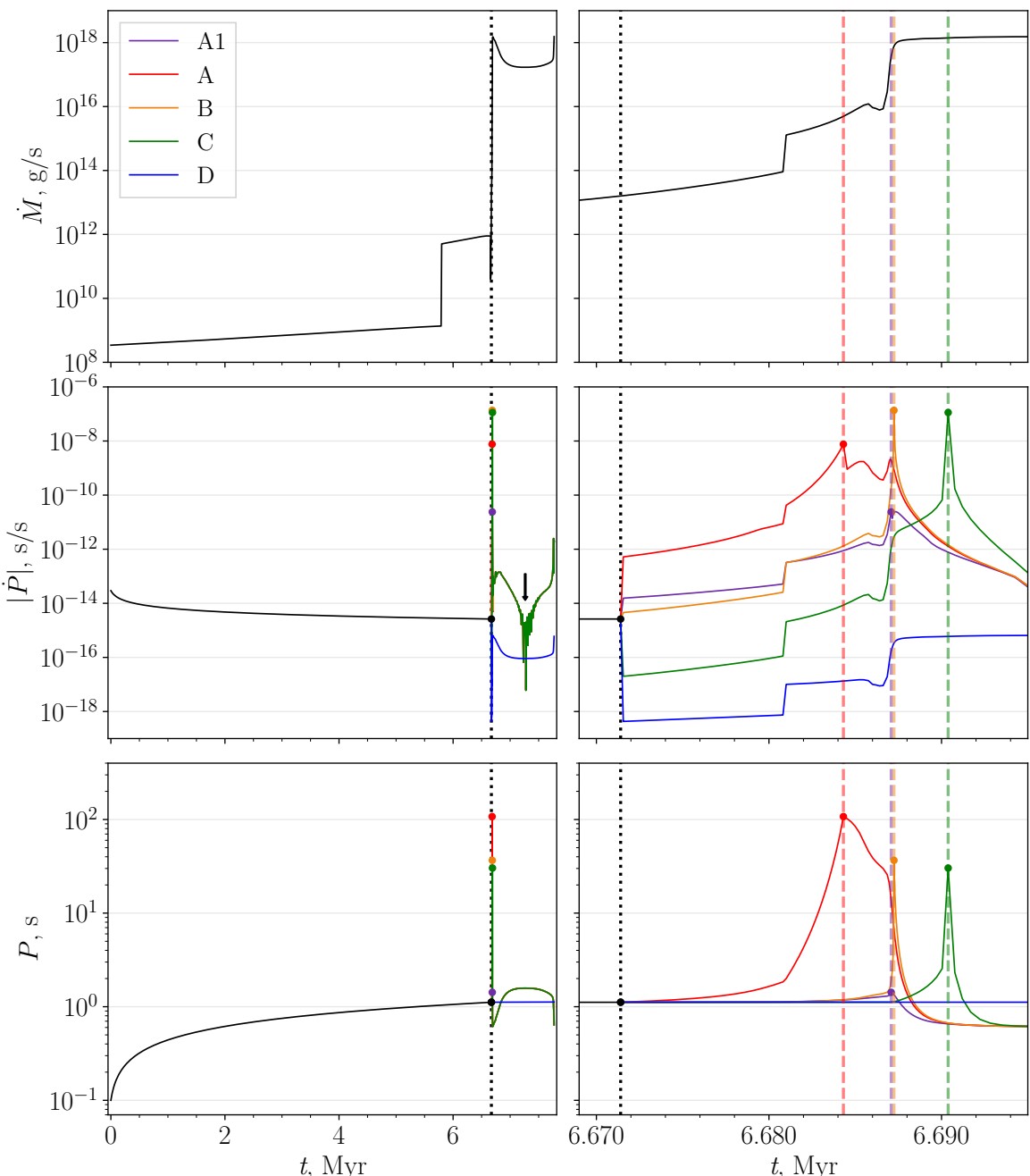

**Figure 2.** Evolution of $\dot{M}$, spin period $P$, and the absolute value of its time derivative $\dot{P}$ for the NS with a constant magnetic field $B = 10^{12}$ G and initial spin period $P_0 = 100$ ms in a binary with the semi-major axis $a = 1280\,R_\odot$. For each propeller model, the NS makes a transition from the ejector to the propeller and then to the accretor stage, except for the model D where the accretor stage is not reached. Black solid curves refer to all propeller models. Black dotted vertical lines and black-filled circles show the transition to the propeller stage, which is the same for all models. Panels on the right side, show the zoomed region near the ejector-propeller transition. Colored vertical dashed lines and colored-filled circles indicate the transition to the accretor stage for each model. The dashed lines are not shown on the left panels, since there they would overlap with the black dotted line. Colored curves for models A1, A, B, and C overlap at the accretor stage and are shown in green on the left panels. The black arrow in the left middle panel points to the sign change of $\dot{P}$ value at 6.7 Myr.

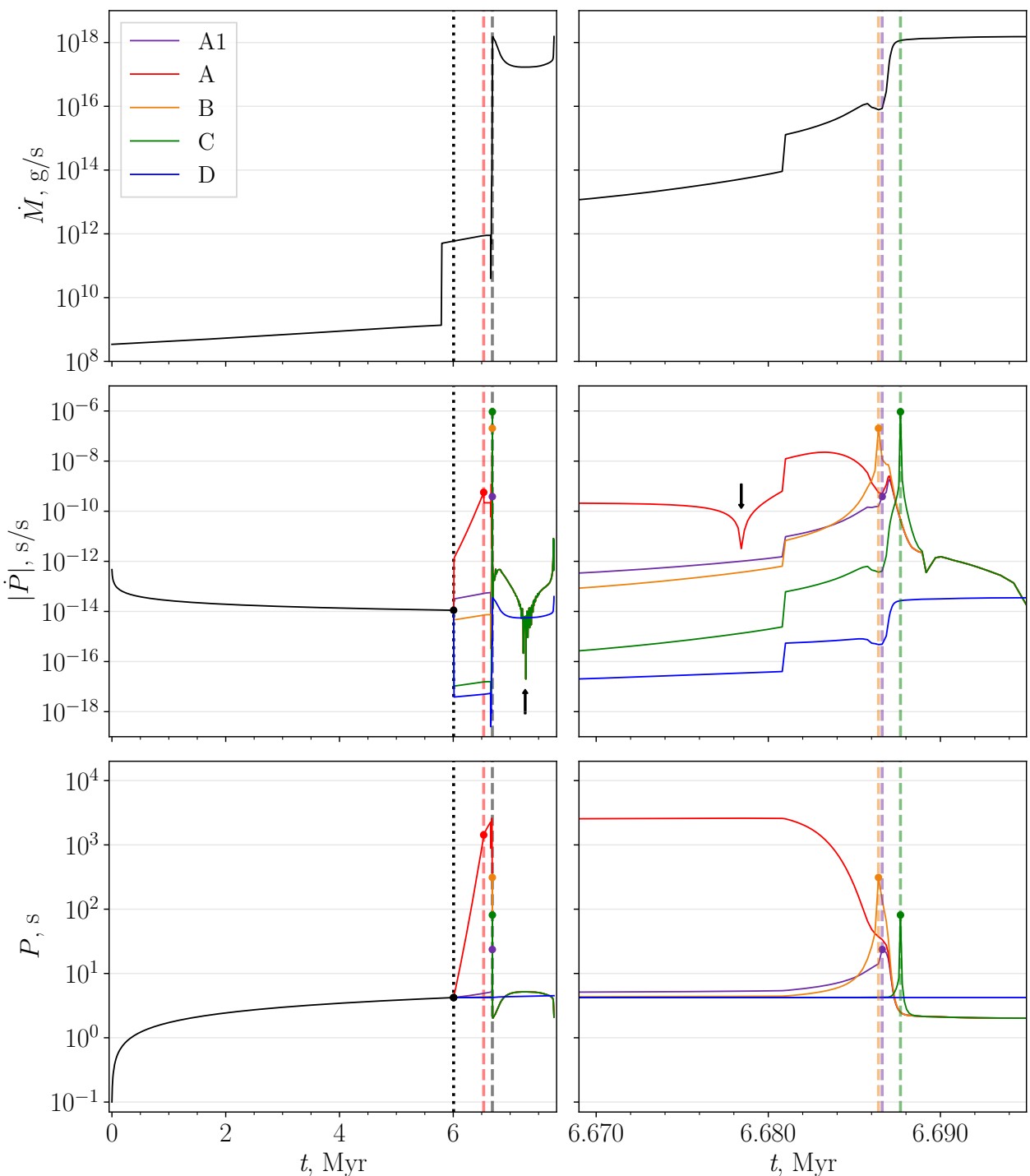

**Figure 3.** Evolution of $\dot{M}$, spin period $P$, and the absolute value of its time derivative $\dot{P}$ for the NS with a constant magnetic field $B = 4 \times 10^{12}$ G. Line styles, points, and colors are the same as in Figure 2. The transition to the accretor stage is now shown in the left panels. Here the grey vertical dashed line shows the transition to the accretor stage for models A1, B, and C, while the red one is for propeller model A. The evolution of an NS within model D does not lead to the accretor stage. In the middle panels, the black arrow at 6.7 Myr on the left and 6.679 Myr on the right show the change in sign of $\dot{P}$. After the NS period reaches $P_{\mathrm{eq}}$, the value of $\dot{P}$ fluctuates visibly.

After reaching the propeller stage, all NSs spin down at a rate that is highly dependent on the propeller model. The difference between the models can be seen well in the zoomed region of Figure 2. The efficiency of this deceleration decreases from model A to model

D, while the spin-down rate of model A1 is mostly of the same order as that of model B. The shape of the $\dot{P}(t)$ generally follows the shape of $\dot{M}(t)$, since the spin-down for each propeller model depends strongly on $\dot{M}$, and among other parameters ($P$, $v$, and $B$), the accretion rate varies the most. Except for model D, the propeller stage for $B = 10^{12}$ G corresponds to the sharp increase of $\dot{M}$ at $\sim$6.7 Myr, well visible in the top left panels of Figures 2 and 3, and therefore lasts just for less than 20,000 yrs for $B = 10^{12}$ G. The duration of the propeller stage of NSs with $B = 4 \times 10^{12}$ G is higher, only because the ejector-propeller transition occurs earlier. For both magnetic field values, the further transition to accretion corresponds to this peak in $\dot{M}$ at 6.7 Myr.

The spin period within the propeller model A grows at a very high rate. The evolution of the NS with $4 \times 10^{12}$ G shows that even for moderate values of $\dot{M} \sim 10^{12}$ g s$^{-1}$ this spin-down is effective enough to bring the NS to the stage of accretion significantly earlier than $\dot{M}$ reaches its maximum $\sim 10^{18}$ g s$^{-1}$. The NS with $B = 10^{12}$ G also starts to accrete before this maximum. As for models A1 and B, $\dot{P}$ is approximately an order of magnitude smaller than for model A. In these cases, for both magnetic field values, the propeller-accretor transition is not due to spin-down, but due to an increase in $\dot{M}$ from $10^{16}$ to $10^{18}$ g s$^{-1}$. The deceleration in model C is less effective, so to reach the transition period $P_{\mathrm{PA}}$ the NS must evolve an additional $\sim$2–3 thousand years with high external pressure at the accretion rate $\dot{M} \sim 10^{18}$ g s$^{-1}$, i.e., very close to the maximum value. Finally, the evolution according to model D is too slow to produce an accretor for both magnetic field values.

An NS can make a transition to the accretor stage either due to a spin-down (models A and C) or due to a rapid increase in $\dot{M}$ (models A1 and B). After the transition, the spin period may decrease or increase depending on the relationship between $K_{\mathrm{su}}$ and $K_{\mathrm{sd}}$ and on the accretion regime—spherical or disc. For NSs with $10^{12}$ G, the disc is formed immediately after the onset of accretion. The transition period $P_{\mathrm{PA}}$ is larger than the equilibrium value $P_{\mathrm{eq}}$ for disc accretion (Equation (16)). That's why, these NSs start to spin up. At the beginning of the accretion, the value of the spin-up torque $K_{\mathrm{su}}$ is significantly larger than the decelerating one. So, the evolution of $\dot{P}$ depends strongly on the accelerating moment $K_{\mathrm{su}}$, initially. Due to the dependence of the accelerating moment on the accretion rate—$K_{\mathrm{su}} \propto \dot{M}^{6/7}$—the $\dot{P}(t)$ curve for model A resembles $\dot{M}(t)$ at the beginning of accretion, as can be seen in the middle right panel of Figure 2.

For NSs with the higher magnetic field value, the accretion stage starts in a similar way, except for model A. Within this model, the transition occurs before $\dot{M}$ has reached a value sufficient for the disc formation. So, in model A, the accretion starts without a disc. For spherical accretion under these conditions, the absolute value of $K_{\mathrm{su}}$ is less than the braking torque. Therefore, during the accretor stage, the NS spins down until $\dot{M}$ is high enough that $K_{\mathrm{su}} > K_{\mathrm{sd}}$. At this moment, the value of $\dot{P}$ changes its sign (black arrow in the middle right panel of Figure 3). Then the NS starts to accelerate as $\dot{M}$ increases with time. After the age 6.686 Myr, the accretion rate will be sufficient to form a disc as in any other propeller model.

For both magnetic field values, in a few thousand years after the propeller-accretor transition, all NSs reach the equilibrium regime of disc accretion. This regime starts at $t \lesssim 6.70$ Myr for $B = 10^{12}$ G and at $t \lesssim 6.69$ Myr for $B = 4 \times 10^{12}$ G. The equilibrium period depends on the magnetic field and the accretion rate, so for every propeller model, the curves tend to saturate at the same value $P_{\mathrm{eq}}$. The spin period obtained by integrating the Euler equation oscillates slightly near the equilibrium value. To make our calculations more precise, we replace the calculated curves $P(t)$ with $P_{\mathrm{eq}}$ and $\dot{P}$ with $\dot{P}_{\mathrm{eq}} = -3P_{\mathrm{eq}}/(7\dot{M})\, d\dot{M}/dt$ from the beginning of this regime. Here $d\dot{M}/dt$ is calculated numerically, which still leads to fluctuations in the calculated values of $\dot{P}$, because $\dot{M}(t)$ is a tabulated curve. These fluctuations in $\dot{P}$ are visible in the middle panels of Figures 2 and 3 after the equilibrium is reached. The spin period $P = P_{\mathrm{eq}}$ varies only because of changes in $\dot{M}$. At the onset of the equilibrium disc accretion after $\dot{M}$ peaks at 6.7 Myr, the NSs start to spin down again. At 7.3 Myr $P_{\mathrm{eq}}$ reaches its maximum value because $\dot{M}$ is at its minimum. Here $\dot{P}$ changes its sign, which is shown as a V-shaped feature and indicated by a black

arrow in the left middle panels of Figures 2 and 3. The subsequent spin-up phase begins at 7.3 Myr, when the accretion rate starts to increase again.

### 4.2. Evolutionary Stages of NSs in a Wide Range of the Magnetic Field

Figure 4 illustrates the evolution of NSs with various magnetic field values within the propeller models C (left panel) and D (right panel). The other parameters remain the same as in the previous calculations in Section 4.1: $P_0 = 100$ ms, $a = 1280\,R_\odot$. We are mainly interested in the evolutionary stages of NSs when the donor star is an RSG, i.e., when the age of the NS is $t \gtrsim 6.7$ Myr. Model C is a representative example of the models in which the NS starts to accrete. Model D represents the case where the NS with the standard magnetic field value is not entering the accretion stage.

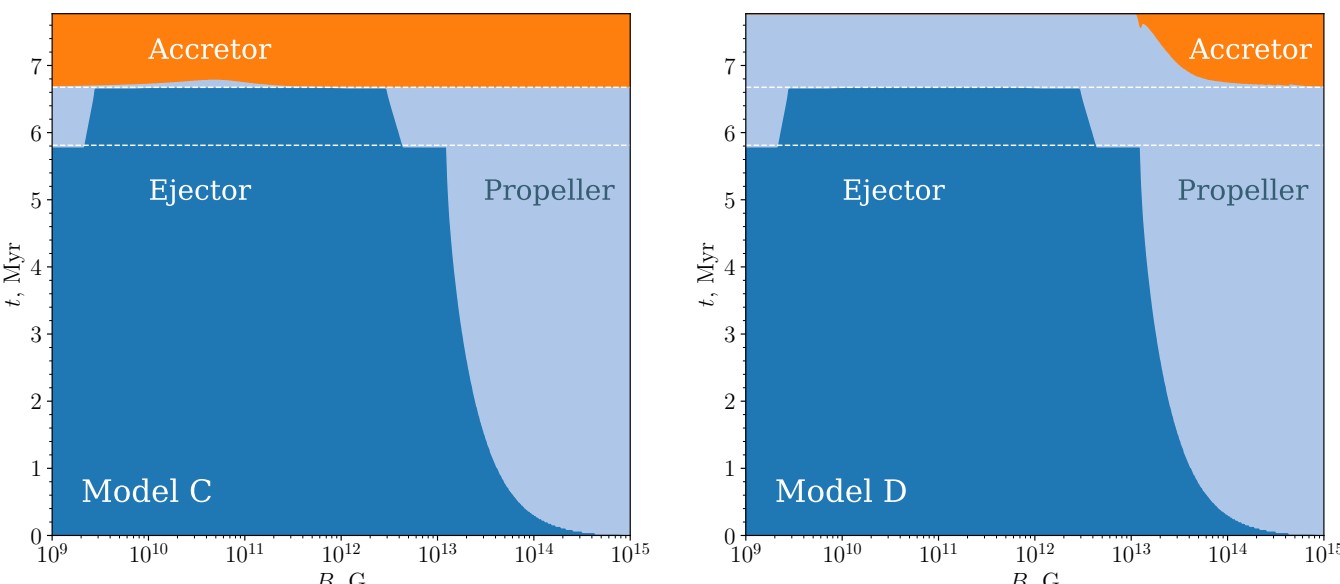

**Figure 4.** Evolutionary stages of NSs with $P_0 = 100$ ms, $a = 1280\,R_\odot$ over the age $t$ of the NS and its magnetic field $B$. Propeller model C is shown on the left panel and model D is shown on the right. Three evolutionary stages are shown in color: ejector (blue), propeller (light blue), and accretor (orange). White dashed lines correspond to the evolutionary phases of the donor star, as in Figure 1.

All NSs start their evolution as ejectors because the stellar wind of the donor is very weak when the NS is born. The ejector phase of evolution is described in the same way for all NSs. The increase of the spin period is insignificant for $B \lesssim 3 \times 10^{12}$ G as the transition to the propeller stage occurs due to the evolution of the donor. For extremely low values of $B \lesssim 2 \times 10^9$ G, the ejector stage ends at 5.8 Myr, while the donor is still in the Main sequence. The first jump in $\dot{M}$ immediately leads to $P_{EP}$ that is short enough to have $P(t) \gtrsim P_{EP}$. NSs at the adjacent slope section of the graph with $B \sim (2 \times 10^9 - 3 \times 10^9)$ G show the same evolution with just one difference—at the ejector-propeller transition $P_{EP}$ is slightly higher, so it is necessary that $\dot{M}$ grows slightly more for the higher $B$ to make $P_{EP} = P(t)$. For NSs with $B$ between $3 \times 10^9$ G and $3 \times 10^{12}$ G the transition to the propeller stage occurs with the second component approaching the RGB. For NSs in this range of $B$ the transition is similar to the previously described transition of the NS from Figure 2 with $B = 10^{12}$ G. For an NS with a higher magnetic field value $B \gtrsim 3 \times 10^{12}$ G the spin evolution starts to influence the onset of the propeller regime. So, the NS with $3 \times 10^{12}$ G $\leq B \leq 2 \times 10^{13}$ G spins down enough that its spin period becomes $P(t) = P_{EP}$ while $\dot{M}$ is only $\sim 10^{12}$ g s$^{-1}$. The evolution of NSs with higher magnetic field values is so fast that the ejector stage ends even before the first jump in $\dot{M}$. So, in this case, the evolution of the donor has almost no effect on the NS evolution.

The area of the propeller stage in Figure 4 is different for the two models, C and D. In model C, before $\dot{M}$ peaks at 6.7 Myr, the spin-down at the propeller stage is negligible. But after $\dot{M}$ reaches its maximum value $\sim 10^{18}$ g s$^{-1}$, the propeller spin-down is always sufficient to bring an NS to the onset of accretion which starts at $t \approx 6.7$ Myr for all values of $B$. Here we can see how strongly the propeller spin-down rate depends on $\dot{M}$, and so $B$ is almost unimportant. While the propeller stage with $\dot{M} \lesssim 10^{12}$ g s$^{-1}$ takes $\sim 1$ million years or more (this can be seen for $B \lesssim 3 \times 10^9$ G and $B \gtrsim 10^{13}$ G), while for $\dot{M} \sim 10^{18}$ g s$^{-1}$ only a few thousand years is enough to reach the accretor stage ($B \sim 10^{10}$–$10^{12}$ G). In the left panel of Figure 4, for values of $B$ near $10^{11}$ G, there is a non-monotonic dependence of the time of the transition from propeller to accretor $t_{\rm PA}$ while the time of the ejector-propeller transition remains constant. This is because within model C the spin-down rate at the propeller stage is $\dot{P} \propto P^3 B^{-4/9}$. At the very beginning of this regime $P = P_{\rm EP}$, which is longer for higher $B$. The interplay between the dependence of $\dot{P}$ on $P_{\rm EP}$ and of $\dot{P}$ on $B$ produces a maximum in $t_{\rm PA}$.

Generally, the spin-down in model D is the least effective one. However, the spin-down in model D can be more effective than in model C under some conditions. The ratio of the braking torque in model D and model C is $K_{\rm D}/K_{\rm C} \propto v^2/v_{\rm ff}^2(R_{\rm m}) \propto R_{\rm m}/R_{\rm G}$, and this ratio is $\sim 1$–40 at $t < 6.7$ Myr for $B \gtrsim 3 \times 10^{13}$ G. Within these conditions $K_{\rm D}/K_{\rm C} \gtrsim 1$, so the spin-down in model D is higher. However, this factor does not make a significant difference, because the spin period evolution with $\dot{M} \lesssim 10^{12}$ g s$^{-1}$, which is $\dot{M}$ value before 6.7 Myr peak, is very slow. So for both models C and D, at the propeller stage, $P$ remains almost the same until the donor becomes an RSG. After 6.7 Myr $K_{\rm D}/K_{\rm C} \sim 10^{-4}$–$10^{-3}$. Therefore, in contrast to model C, in model D it takes significantly longer for an NS to begin to accrete.

In the right panel of Figure 4, at later stages of the donor evolution (above the second dashed line), the border between the propeller and the accretor stage, $t_{\rm PA}(B)$, is curved. i.e., in this case, the magnetic field value influences the time of the propeller-accretor transition. The spin-down within this model does not depend directly on $B$, but it depends on the spin period $\dot{P} \propto P^3$. Since all NSs start as ejectors, at the beginning of the propeller regime the spin-down is initially much higher for higher values of $B$ as $P_{\rm EP} \propto B^{1/2}$. The transition period $P_{\rm PA} \propto B^{2/3}$ also depends on $B$, but this dependence is weaker than $\dot{P}(B) \propto B^{3/2}$ and does not change $t_{\rm PA}(B)$ much. Thus, due to the lower initial value of the spin period at the onset of the propeller stage, NSs with lower magnetic fields start to accrete later, and NSs with $B \lesssim 10^{13}$ do not have enough time to reach the accretor stage at all.

The shape of the accretor stage region in the right panel of Figure 4 has a feature at $B \approx 2 \times 10^{13}$ G. Near this value $t_{\rm PA}(B)$ is non-monotonic. Mainly, this feature is related to the fact that the transition ejector-propeller happens due to the increase of $\dot{M}$. This means that for lower fields it occurs for shorter periods. Spin-down at the propeller stage in model D is very ineffective. So, the transition propeller-accretor again happens due to the increase in $\dot{M}$ and the spin-period is nearly equal to the one at the ejector-propeller transition. As $P_{\rm PA} \propto (B^2/\dot{M}v^2)^{1/3}$, $P_{\rm EP} \propto B^2$, and $\dot{M}v^2$ increases at the latest stages that we consider, there is a complicated interplay between the parameters resulting in the non-monotonic behavior visible in the Figure.

In Figure 4 we clearly see that the evolution of the donor is the main driver of the NS evolution for a wide range of magnetic fields (including standard fields $B \sim 10^{10}$ few $\times 10^{12}$ G), except some boundary cases discussed above. This justifies that we do not consider various values of the age difference between the NS and the donor. The evolution of the donor remains the same for any value of this parameter and influences the evolution of the NS in almost the same way.

## 5. Discussion

### 5.1. Magnetar Evolution

In our model, we neglect the magnetic field decay. In particular, we ignore the possibility that the initial field could be high and then rapidly decay. It is expected that after a few e-foldings, the field is saturated at some value ($\sim 1/20$ of the initial field) that

corresponds to the Hall attractor stage [38]. Effectively, on the scale of our problem, rapid decay of the magnetar scale field with subsequent saturation results just in a larger initial spin period ∼10 s, i.e., a typical magnetar value. In this subsection, we calculate how this could influence our conclusions.

In Figure 5 we present evolution of an NS with $P_0 = 10$ s and constant field $B = 4 \times 10^{12}$ G. All notations are the same as in Figures 2 and 3.

In this case, the NS is born already at the propeller stage. i.e., this means that if we have in mind a magnetar scale initial magnetic field and short spin period then the transition ejector-propeller happens very quickly in the history of the compact object due to the rapid spin-down. However, as soon as the mass loss by the donor starts to increase, the NS (in the cases of models A, A1, B, and C) becomes an accretor approximately as fast as in Figure 3. i.e., the duration of the propeller stage relative to the accretor stage at high $\dot{M}$ is very similar to the case of a standard (short) initial spin period. In the case of model D, as in Figure 3, the NS remains at the propeller stage.

We conclude that large initial spin periods $P{\sim}10$ s (mimicking magnetar fields with rapid decay and Hall attractor) do not influence the main results of our analysis.

### 5.2. Relative Spin-Down Rates for Ejectors and Different Propeller Regimes

In [46] the authors claimed that for a very wide range of realistic parameters, spin-down at the propeller stage is more efficient than at the ejector stage. Figures 2 and 3 demonstrate that for our parameters this is not the case for models C and D. Even for model B there is an interval when $\dot{P}$ at the propeller is smaller than at the ejector stage.

We think that basically, this claim assumed that $R_{\mathrm{m}} = R_{\mathrm{A}} \approx R_{\mathrm{co}}$. This can be true if the density profile in the envelope is close to the one for the free falling matter and if the NS is close to the propeller-accretor transition. However, if, according to the suggestion by Davies and Pringle [43], $R_{\mathrm{m}} = R_{\mathrm{A}}^{7/9} R_{\mathrm{G}}^{2/9} \gg R_{\mathrm{A}}$ then the propeller spin-down can be less effective than the ejector spin-down rate, in correspondence with our results.

In our study, we also neglect the phase of the 'very rapid rotator', see [43]. In this regime when $R_{\mathrm{m}} \approx R_{\mathrm{l}}$, the spin-down rate, according to Davies and Pringle, can be even lower. This phase is still a hypothetical one. Its existence is not supported by studies of NS evolution in binaries. Finally, if evolution is mainly driven by the increase of $\dot{M}$ (as it is in the case we analyze), not by the spin-down, then this regime can be neglected.

### 5.3. Disc Formation at the Propeller Stage

In our study, we account for the possibility of disc formation at the accretor stage. However, a disc can be formed already at the propeller stage. We neglect this possibility in all our models. Disc formation might result in a reduction of the magnetospheric radius. Thus, on the one hand, our deceleration moment can be lower than in the case of disc formation (see e.g., ref. [47] about magnetosphere-disc interaction at the propeller stage). On the other hand, the critical period for the transition can be shorter than we consider. Both effects work in one direction: they shorten the propeller stage. i.e., our conclusions regarding models A, A1, B, and C are conservative in this respect.

However, we have to note that in many cases, the beginning of accretion is related not to a gradual spin-down, but to a rapid increase of $\dot{M}$ due to enhanced mass loss by the donor. In such cases, we expect that the disc appearance would not significantly modify our results even quantitatively. Still, this question deserves a more detailed analysis, especially for model D, which is beyond the scope of this paper as most probably it requires detailed numerical modeling of the interaction between the magnetic field and surrounding medium.

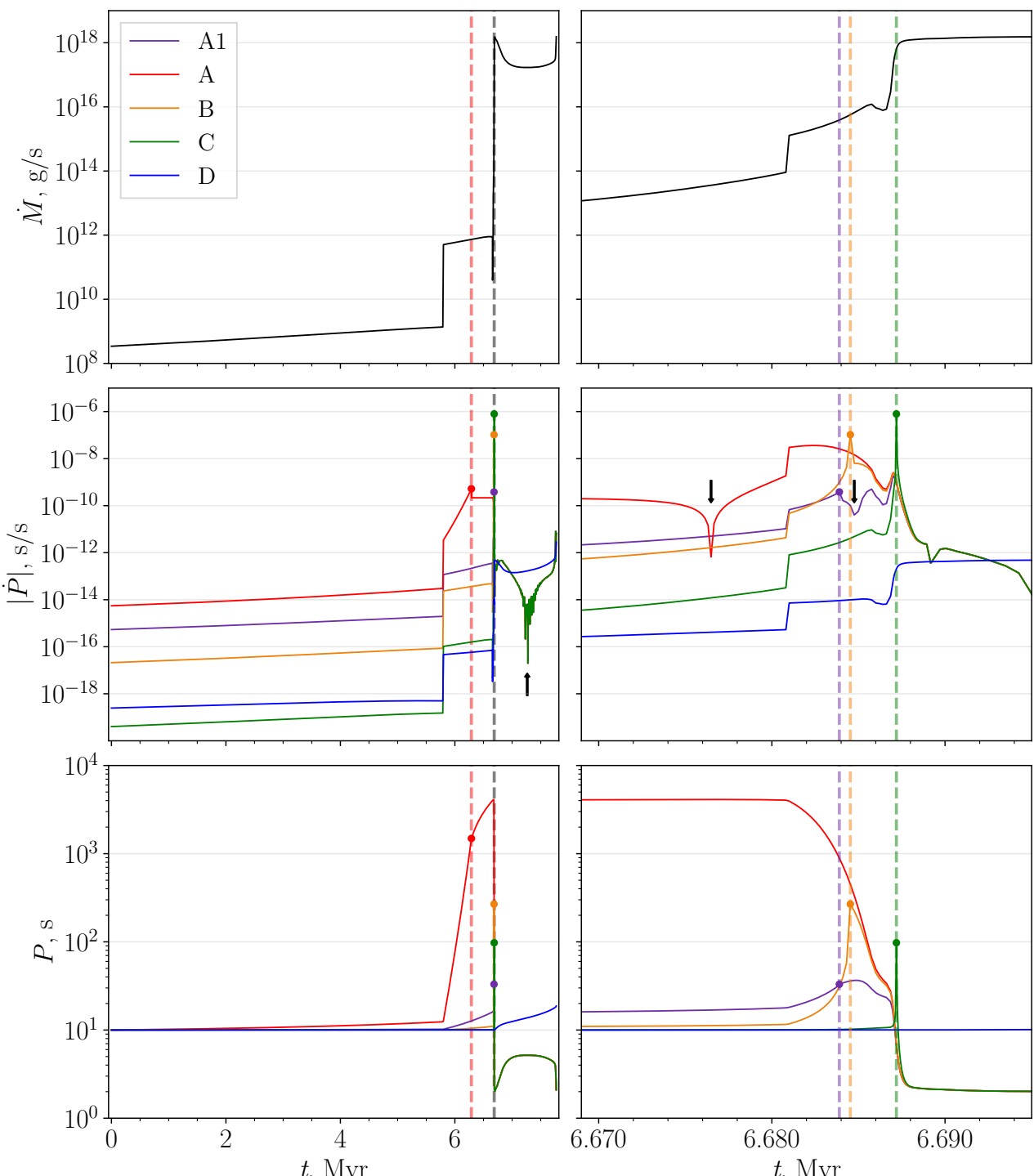

**Figure 5.** The evolution of an NS with $P_0 = 10$ s, $B = 4 \times 10^{12}$ G in a binary with $a = 1280\,R_\odot$. The first evolutionary stage of the NS is the propeller stage. Therefore there is no vertical dotted line for the ejector-propeller transition. Otherwise, all curves, lines, and dots have the same style as in Figure 3. Black arrows indicate changes in the sign of $\dot{P}$ as in Figures 2 and 3. In the middle right panel, the curve for model A1 has an additional sign change at 6.685 Myr after the propeller-accretor transition, similar to the model A (6.676 Myr), which is also indicated by a black arrow.

### 5.4. Wind Accretion Flow in a Red Supergiant Binary

In this study, we assumed a usual simplified description of the wind accretion, see e.g., refs. [32,48]. However, the realistic situation can be much more sophisticated. This is especially true for the systems with massive evolved donors, see [49] and references therein.

Wind parameters of massive evolved stars can demonstrate a very complicated behavior including intense pulsations. The flow can significantly deviate from the spherical symmetry and wind velocity can vary in a complicated way depending on the distance from the donor, not following the $\beta$-law that we apply.

We selected the size of the binary system sufficiently large to avoid the Roche lobe overflow. However, when the radius of the donor is about the Roche lobe radius, a new regime of mass transfer can be realized [50]. We do not account for this possibility and so underestimate the accretion rate for smaller values of the orbital separations.

In addition, the accretion flow in the vicinity of the compact object might have a complicated topology, e.g., ref. [51]. Thus, the exact amount of matter gravitationally captured by the NS, its angular momentum, and its dependence on time can deviate from the values we assume in our modeling. Conditions for disc formation also can be different from those that we use. Precise modeling of a particular system might be based on realistic 3D calculations for the actual parameters of the binary. Still, at the moment, the properties of SWIFT J0850.8-4219 are not precisely known. So, a more simplified analysis presented above is justified.

### 5.5. The Corotation Radius Value and Magnetic Inclination

Usually, the corotation radius is taken in the form of Equation (6). However, recently it was admitted [40] that even in the case of an aligned rotator without a disc formation, the equatorial value of the centrifugal barrier is $0.87R_{co}$, where $R_{co}$ is defined by Equation (6). We checked if this modification could influence our results. With a smaller value of $R_{co}$, we obtained slightly higher values of $P_{eq}$ at the accretor stage, but in general, it did not change our main results and conclusions.

The situation with the centrifugal barrier becomes more complicated if the magnetic inclination and a realistic magnetosphere structure are taken into account. In all our calculations we neglect that surfaces corresponding to the critical radii ($R_{co}$, $R_m$, $R_{Sh}$) are not spheres. Also, we do not account for the magnetic inclination, i.e., for the misalignment between spin and magnetic axis.

Accounting for these details might result in slight modifications of spin-up/down torques, in the value of the equilibrium period, and in the accretion luminosity as the fraction of matter that reaches the NS surface might be different for various inclinations. Still, it is reasonable to assume that our general conclusions might stay intact.

### 5.6. Luminosity at the Propeller Stage

One of the arguments in favor of the propeller interpretation of the system SWIFT J0850.8-4219 is related to its low X-ray luminosity $L \approx 4 \times 10^{35}$ erg s$^{-1}$ [22]. At the accretor stage, assuming a realistic binary separation and properties of the secondary component, this value is expected to be much higher $L_A \sim 10^{37}$ erg s$^{-1}$. In the propeller regime, the matter flow is stopped by a rapidly rotating magnetosphere. Thus, just a tiny fraction of matter can reach the NS surface. We can determine the relative accretion efficiency as $\zeta = L/L_A$. According to [22] this parameter is $\zeta \sim 10^{-2}$.

Let us estimate the relative accretion efficiency expected in our model. First, we consider model D. This is the only scenario that results in a long propeller stage in a binary with the NS in the closest possible orbit $a = 1280\,R_\odot$. As can be seen in Figures 2, 3 and 5 when the donor is an RSG the value of the accretion rate is $\dot{M} \approx 2 \times 10^{17}$ g s$^{-1}$. So, the accretion luminosity would be $L_A = GM\dot{M}/R \approx 4 \times 10^{37}$ erg s$^{-1}$. From this we obtain $\zeta \approx 10^{-2}$. This is consistent with the expected accretion efficiency suggested in the original paper [22] and seems realistic.

If we consider a slightly more effective propeller spin-down, model C, we need to increase the separation $a$ up to at least 3950 $R_\odot$ in order to produce a long propeller stage (otherwise, the probability to detect one of just two known systems at this stage is negligibly small). In this case, the rate of matter capturing is lower $\dot{M} \approx 7 \times 10^{15}$ g s$^{-1}$, so $L_A \approx 10^{36}$ erg s$^{-1}$ and the efficiency is higher $\zeta \approx 0.3$. This value seems to be too high for the propeller regime. From this, we can conclude that the propeller model C does not describe the system SWIFT J0850.8-4219 well even if the duration of the propeller stage is made long. The remaining propeller models A1, A, and B require even higher values of the semi-major axis $a$, which results in a higher efficiency $\zeta$. Therefore, these propeller models appear less realistic than model D in the context of this binary within our scenario of evolution.

*5.7. The Possibility of Settling Accretion in SWIFT J0850.8-4219*

Reduced accretion luminosity can be explained also in the model of settling accretion. This accretion regime has been proposed in [44] and later considered in detail in the series of papers [52–54]. This type of accretion can occur if the X-ray luminosity is below the critical value $L_{SA} \sim 10^{36}$ erg s$^{-1}$. It is determined by the characteristic cooling time $t_{cool}$ and the free fall time $t_{ff}$. Otherwise, if $L \gtrsim L_{SA}$ then $t_{cool} \lesssim t_{ff}$, i.e., cooling due to the Compton and radiative processes is effective. In this case, the usual Bondi accretion proceeds.

Generally, the subsonic settling accretion is characterized by the existence of a hot convective shell around the NS magnetosphere. Due to the interchange instability, a small amount of matter enters the magnetosphere. This could explain the low X-ray luminosity in SWIFT J0850.8-4219 in comparison with the maximum value $L_A$. However, our modeling shows that for realistic parameters the accretion stage starts when $\dot{M}$ is already too high and in addition, an accretion disc is formed around the NS which prevents the appearance of the settling accretion regime. This regime can be realized (and lasts long enough) only if the separation between the two components is substantially large $a > 6000\,R_\odot$. The NS circular velocity in such an orbit is only 21 km s$^{-1}$, while typical kick velocities are 100–1000 km s$^{-1}$, see e.g., ref. [55] and references therein. So, after a supernova explosion of the NS progenitor, very fine-tuning would be required to maintain this binary. We therefore conclude that the settling accretion is not a viable option for SWIFT J0850.8-4219.

**6. Conclusions**

In this paper, we applied different models of spin-down aiming to analyze the hypothesis proposed in [22] that the NS in SWIFT J0850.8-4219 is at the propeller stage. We demonstrate that for all but one of the models duration of this stage is too short to provide a significant probability of detection of one of just two known systems as a propeller. Only for the model of the supersonic propeller by Davies and Pringle [43] the NS remains at the propeller stage with a supergiant companion for the parameters suggested for SWIFT J0850.8-4219. Measurements of the NS spin period and its derivative are necessary to confirm if the NS is in the propeller regime and to distinguish between different scenarios of the propeller stage. In general, studies of SyXBs might be fruitful in understanding of the NS behavior at the propeller stage.

**Author Contributions:** The authors contributed to this study equally. Conceptualization, S.B.P.; methodology: M.D.A. and S.B.P.; software: M.D.A.; writing—original draft preparation, M.D.A. and S.B.P.; writing—review and editing, M.D.A. and S.B.P.; supervision, S.B.P. All authors have read and agreed to the published version of the manuscript.

**Funding:** M.A. acknowledges support from the Basis Foundation grant 23-2-1-74-1. S.P. acknowledges support from the Simons Foundation which made possible the visit to the ICTP.

**Data Availability Statement:** The code used for modeling is accessible on request. Please, contact the first author.

**Acknowledgments:** We thank all the anonymous referees for their comments and suggestions that helped us to improve the manuscript.

**Conflicts of Interest:** The authors declare no conflicts of interest.

## Abbreviations

The following abbreviations are used in this manuscript:

HMXB　High-mass X-ray binary
NS　　Neutron star
RGB　　Red giant branch
RSG　　Red supergiant
SyXB　Symbiotic X-ray binary

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
