# Peer review of "Probing the Propeller Regime with Symbiotic X-ray Binaries"

_universe, doi:10.3390/universe10050205_

Round 1

Reviewer 1 Report

Comments and Suggestions for Authors

The manuscript studied the propeller effect for the accreting neutron star binary systems, e.g., applied to  SWIFT J0850.8-4219 that contains a neutron star, and it is interesting and worthy of publication.  Some questions should be clarified before it is recommended for publication. 

1. Authors might add some sentences to describe the research history of propeller model in explaining the related phenomena of accretion systems. 

2. Does the magnetic inclination angle between the rotational axis and magnetic pole have contribution to or influence on the propeller effect?

3. The neuron star co-rotational radius, magnetosphere radius and Alfven radius are similar while the propeller happens, please present some explanation in detail. 

4. For the different stages of accretion from HMXB to LMXB, the magnetic field of neutron star decays by accreting mass (the magnetosphere radius shrinks with accretion), while the stellar spin period is spin-up, how do we evaluate the propeller phenomena for the different accretion environments? 

Comments on the Quality of English Language

It is perfect.

Author Response

We thank the referee for the useful comments that helped to improve the manuscript. 
We took into account all the comments and suggestions. 
Also, we expanded the Discussion part (sec 5). 
Changes are marked with the bold font.

>1. Authors might add some sentences to describe the research history of propeller model in explaining the related phenomena of accretion systems. 

In the Introduction, we added several sentences with historical references to early studies and references to new results related to observations of accretor-propeller transitions in different binary systems.

>2. Does the magnetic inclination angle between the rotational axis and magnetic pole have contribution to or influence on the propeller effect?

Yes, we added a discussion in Sec. 5.5. 

>3. The neuron star co-rotational radius, magnetosphere radius and Alfven radius are similar while the propeller happens, please present some explanation in detail. 

We assume that this comment is related to Sec 5.2. We added additional phrases in this section to make our statement clear.

>4. For the different stages of accretion from HMXB to LMXB, the magnetic field of neutron star decays by accreting mass (the magnetosphere radius shrinks with accretion), while the stellar spin period is spin-up, how do we evaluate the propeller phenomena for the different accretion environments? 

We added a sentence (and a reference to Konar, Bhattacharya 1997) in the Section where we discuss the field decay. 
As in the case of SWIFT 0850.8-4219 we are dealing with a system with a massive donor that does not fill the Roche lobe, we can safely neglect the field decay due to accretion. 

Reviewer 2 Report

Comments and Suggestions for Authors

This paper provides evolutions of the mass accretion onto a magnetized neutron star, taking account of an evolution of the donner star. The author discusses duration of propeller stage and the detection probability of the X-ray binary with red supergiant as donner at the propeller stage.  

  Finally, they suggest the importance of the determination of the NS properties, such as the spin period and its derivative.  But readers may hope to know more specific expected results from the further observation.

There are some specific comments.

L15: V. Shvartzman should be Shvartsman.

L52: Need the explanation of 

L179: Why does authors deal with the relativistic particle? Are the wind particles relativistic?  

Figure 2:  I find only one black arrow in the figure.  There are several periods decreases around 6690, which I can see in the right panels.  It is very confusing. 

Figure 3:  The same comments as for figure 2.

L478: What do the authors mean 'the different possibilities'?  The accretion stages? or the models?

Author Response

We thank the referee for the useful comments that helped to improve the manuscript. 
We took into account all the comments and suggestions. 
Also, we expanded the Discussion part (sec 5). 
Changes are marked with the bold font.

>L15: V. Shvartzman should be Shvartsman.

Corrected

>L52: Need the explanation of ’’. 

The explanation is added.

>L179: Why does authors deal with the relativistic particle? Are the wind particles relativistic?

We re-phrased this sentence. Originally, we intended to say that as the relativistic wind (at the ejector stage) cannot prevent the external matter from entering the light cylinder then the NS makes transition to the propeller stage where the relativistic wind is quenched. But it seems that we made our statement in an unclear way. So, now the text is modified appropriately.

>Figure 2:  I find only one black arrow in the figure.  There are several periods decreases around 6690, which I can see in the right panels.  It is very confusing. 

Corrected.

>Figure 3:  The same comments as for figure 2.

Corrected

>L478: What do the authors mean 'the different possibilities'?  The accretion stages? or the models?

We added a description of the possibilities.

Reviewer 3 Report

Comments and Suggestions for Authors

This is an interesting paper. However it might have limited audience due to the specialized content. Nevertheless it is worth publishing this work.

Author Response

We thank the referee for the report. 

>This is an interesting paper. However it might have limited audience due to the specialized content. >Nevertheless it is worth publishing this work.

We agree that the paper can be interesting for a limited audience as it addresses a particular problem concerning one particular system. In the revised version we slightly expanded the Introduction and Discussion. This, hopefully, can make the paper interesting for a little bit wider circle of researchers.